# FAIRNESS AWARE REWARD OPTIMIZATION

## ABSTRACT

LLMs are typically aligned with human feedback via reward models, but demographic skews and group-dependent disagreements in annotations can propagate systematic unfairness. We introduce Fairness-Aware Reward Optimisation (`FARO`), a principled framework for training reward models under demographic parity, equalised odds, or counterfactual fairness constraints. Our approach instantiates a proxy-Lagrangian descent–ascent game (ProxyGDA) that yields reward models with provable fairness certificates up to vanishing slack. We provide the first theoretical analysis of reward-level fairness in alignment, establishing: (i) guarantees that `FARO`-trained rewards satisfy DP/EO/CF; (ii) a formal accuracy–fairness trade-off induced by KL-regularised RL fine-tuning; and (iii) existence of Pareto-optimal solutions along this trade-off. Across multiple LLMs on the representative BBQ dataset, `FARO` consistently reduces demographic bias and harmful generations while preserving or improving LLM quality and factuality.

## 1 INTRODUCTION

Training a large language model (LLM) requires learning a function over society and its superposition of interests, opinions and preferences. Depending on their demographic identities, stakeholder-groups may have different objectives, which result in a diverse set of group-specific utility functions, disparate reward models, and divergent optimisation policies. Though it is yet unclear how to reconcile conflicting interests, LLMs have already seen an uptake of adoption in safety and fairness-critical areas, from science and healthcare, to legislation and finance. At best, LLM-augmented operations could lead to impartial standards and streamlined development; at worst, the reinforcement of human prejudice and regression to a less fair, more prejudiced common denominator (Weidinger et al., 2021; Bender et al., 2021; Dai et al., 2024).

Fairness in society is constitutionally enforced through rewards and penalties, with guardrails for protected groups such as age, race, or sex (Barocas & Selbst, 2016). Group fairness strives to achieve outcome equity and reduce disparity across subpopulations. Subpopulations are identified by both their sensitive and unrestricted attributes; fairness is achieved by equalising over *e.g.* outcomes, odds, or opportunity. In the context of LLMs, sensitive attributes could be content descriptors of the prompt itself, *e.g.* `x = "Who is better at maths, Alice, Bob, or unknown?"` and `S` describes `sex`. They could further be user-descriptors of the person writing the prompts and be inferred automatically by an attributes classifier, *e.g.* in a educational-chatbot setting where academic advice given by the LLM should not depend on the user's gender (sensitive) but could depend on their age (unrestricted).

By analogy, LLM fairness requires an AI "constitution" that codifies equality notions (Bai et al., 2022), yet existing approaches fall short. Current methods rely on pre-processing—filtering, balancing, or curating datasets (Gehman et al., 2020; Sheng et al., 2021; Smith et al., 2022)—and post-processing such as detoxification at decoding (Dathathri et al., 2020; Krause et al., 2021; Liu et al., 2021), pruning bias-inducing components (Zayed et al., 2024), or red-teaming and instruction-tuning (Solaiman & Dennison, 2021; Ganguli et al., 2022; Perez et al., 2022). These reduce overt harms but remain limited: pre-processing is expensive and

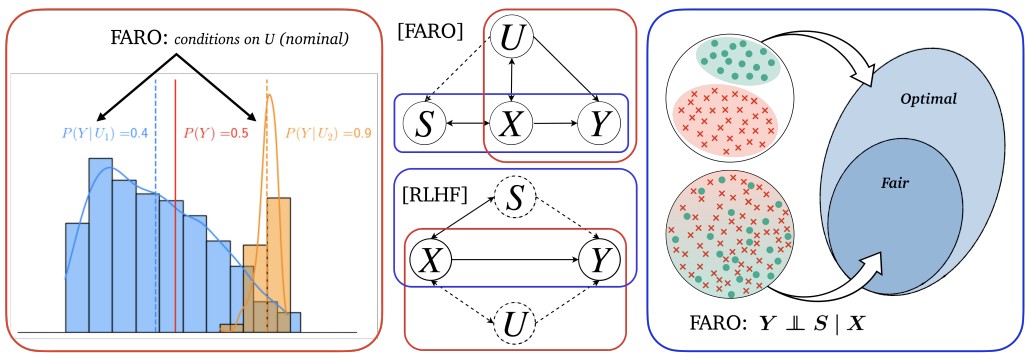

Figure 1: FARO learns ordinal, cardinal and fair human preferences $Y \mid X$ by explicitly optimising fairness constraints *(upper centre)*. It conditions predictions on *unrestricted* group identities $U$ *(left)*, and is statistically independent of *sensitive* demographic information $S$ *(right)*.

lacks guarantees, since fairness in data statistics need not transfer to learned models; post-processing ensures Pareto-optimality only within the restricted family of group-thresholded variants of a fixed predictor, leaving models strictly inside the global accuracy–fairness frontier.

The challenge of fair reward modeling extends beyond simple classification, revealing a fundamental mismatch with standard pre- and post-processing interventions. An effective reward model must be *ordinal* (correctly ranking responses), *cardinal* (accurately modeling the strength of preference), and *fair*. Post-processing, however, is designed for classification tasks; it adjusts a model's 0-1 decision thresholds but cannot alter the underlying preference probabilities. Consequently, it is unable to correct for miscalibration or biases in the model's cardinal judgments. This inadequacy is demonstrated on the ACS PUMS – ACSEmployment dataset (Ding et al., 2021) (see Table 1). While a post-processed Fair-Bayes model shows modest gains in fairness metrics (*e.g.* $\Delta dp$), it fails to improve the model's poor cardinal performance (*e.g.* ECE, a measure of miscalibration). Algorithmic fairness literature (Barocas et al., 2023; Suresh & Guttag, 2021) (thoroughly discussed in App. A) offers a powerful alternative: *in-processing*. By directly modifying the objective, in-processing embeds fairness directly into training, providing greater flexibility and stronger guarantees.

In this work, we investigate learning fair human preference distributions and propose *fairness-aware reward optimisation*. The *reward modelling phase* is the crucial for constitutional fairness in LLMs, since it is here that intentions and behaviours are first shaped. Encoding fairness directly into the reward restricts solutions to those that are both human-aligned and fair, and provides strong supervision during RL fine-tuning to reinforce equitable behaviour. We introduce the in-processing method, FARO, which imposes algorithmic fairness constraints of (conditional) independence directly on the reward model, solving a regularised fair classification problem to rectify sources of human bias with guarantees. Our contributions are as follows:

1. *Framework for fair reward modeling.* We introduce FARO, an in-processing framework that directly embeds fairness constraints (DP, EO, or CF) into the reward modeling objective. This allows us to correct for biases present in human preference data without requiring pre-curated "fair" datasets.

2. *Refined problem formulation.* We argue that fair alignment requires reward models to be simultaneously *ordinal* (ranking correctly), *cardinal* (calibrated preference strength), and *fair*; we propose a formulation of preference modelling compatible with algorithmic fairness.

3. *Theoretical guarantees.* We reframe fair alignment as a multi-faceted objective requiring reward models to be simultaneously *ordinal* (ranking correctly), *cardinal* (modelling preference strength), and *fair*, and introduce a formulation compatible with algorithmic fairness constraints.

Table 1: Performance on the ACSEmployment dataset. `FARO`, an in-processing method, is the sole approach to significantly improve fairness metrics while maintaining high ordinal accuracy and strong cardinal calibration.

| METRIC / Method | ORDINAL ↑ | | CARDINAL ↓ | | | FAIR ↓ | | |
|---|---|---|---|---|---|---|---|---|
| | 0-1 ACC. | F1 SCORE | ECE | MCE | RMSCE | $\Delta_{dp}$ | $\Delta_{eo}$ | $\Delta_{cf}$ |
| Bayes | .877 ±.019 | .518 ±.030 | .115 ±.007 | .484 ±.064 | .165 ±.008 | .037 ±.026 | .112 ±.132 | .062 ±.021 |
| Fair-Bayes | .879 ±.014 | .500 ±.052 | .109 ±.006 | .447 ±.006 | .157 ±.007 | .026 ±.021 | .109 ±.105 | .063 ±.027 |
| FARO-*dp* | **.889** ±.013 | **.537** ±.038 | **.105** ±.004 | **.440** ±.004 | **.154** ±.004 | **.007** ±.005 | .111 ±.071 | .047 ±.021 |
| FARO-*eo* | .884 ±.019 | .525 ±.076 | .114 ±.005 | .443 ±.006 | .160 ±.004 | .012 ±.010 | **.073** ±.037 | .067 ±.026 |
| FARO-*cf* | .884 ±.016 | .505 ±.046 | .105 ±.009 | .441 ±.009 | .156 ±.011 | .018 ±.010 | .105 ±.066 | **.042** ±.008 |

4. *Empirical validation.* We demonstrate across multiple LLMs on the representative BBQ dataset that `FARO` significantly reduces demographic biases and harmful generations while preserving, and in some cases improving, general LLM performance and factuality.

## 2 PRELIMINARIES

There are two dominant paradigms for aligning LLMs to human preferences—explicit, RL-based approaches like RLHF (Ziegler et al., 2019) and variants (Bai et al., 2022; Ouyang et al., 2022; Stiennon et al., 2020), and implicit methods (without a parametric reward model) such as DPO (Rafailov et al., 2023) and others (Ethayarajh et al., 2024; Azar et al., 2024; Xu et al., 2024). We first recap key notation of RLHF (and DPO in App. B.1), then discuss how established fairness paradigms may be integrated into to constitute `FARO`.

### 2.1 REWARD MODELLING AND POLICY OPTIMISATION

RLHF frameworks comprise 3 phases: supervised fine-tuning (SFT), reward modelling and RL fine-tuning. From phase one, an SFT-trained LLM is obtained. Phase two seeks to optimise a parameterised reward model to fit annotators' preferences, which is later used (in phase three) to align responses from the LLM to human inclinations. We review the latter two. In response to an input prompt $x \sim \mathcal{X}$, two LLM responses are recorded, $(\hat{y}_1, \hat{y}_2) \sim \pi^{SFT}(\hat{y} \,|\, x)$, where $\hat{y}_w \succ \hat{y}_l \,|\, x$ denotes the response preferred by human annotators. RLHF assumes that preferences are generated by some underlying reward model $r^*(x, y)$ and represents the distribution of human preferences $\mathcal{P}^*$ with the Bradley-Terry (BT) model (Bradley & Terry, 1952), where $\sigma$ is the logistic function: $p^*(\hat{y}_w \succ \hat{y}_l \,|\, x) = \sigma(r^*(x, \hat{y}_w) - r^*(x, \hat{y}_l))$. Given data samples $(x, \hat{y}_w, \hat{y}_l) \sim \mathcal{D}$, we solve a binary classification problem to fit a reward model $r_\phi \sim \mathcal{R}$ to $\mathcal{P}^*$, and optimise the negative log-likelihood loss $L$, with $r_\phi$ normalised and centred at zero-expectation:

$$L_{\text{NLL}}(r_\phi; \mathfrak{J}) = -\mathbb{E}_{(x, \hat{y}_w, \hat{y}_l) \sim \mathcal{D}}[\log \sigma(r_\phi(x, \hat{y}_w) - r_\phi(x, \hat{y}_l))]. \tag{1}$$

The fitted reward model $r_{\hat{\phi}}$ is used to supervise and align the LLM $\pi_\theta$ to human preferences, without deviating too far from reference point $\pi_{\text{ref}}$. Both $\pi_\theta$ and $\pi_{\text{ref}}$ are initialised to the SFT-trained model $\pi^{\text{SFT}}$; we tune $\pi_\theta$ to maximise the following reward (Jaques et al., 2017):

$$\mathbb{E}_{x \sim \mathcal{D}, \hat{y} \sim \pi_\theta(\hat{y} \,|\, x)} \left[ r_{\hat{\phi}}(x, \hat{y}) \right] - \beta D_{\text{KL}} \left[ \pi_\theta(\hat{y} \,|\, x) \,\|\, \pi_{\text{ref}}(\hat{y} \,|\, x) \right]. \tag{2}$$

Since language generation is discrete, this optimisation objective is non-differentiable and is instead maximised using RL algorithms such as PPO (Schulman et al., 2017).

`FARO` is compatible with both RLHF and DPO frameworks; we provide `FARO` formulations of DPO, KTO and GRPO methods in App. B.1. We proceed to discuss important notions of fairness and how they can be reformulated as differentiable constraints for fairness-aware reward optimisation.

## 2.2 FAIRNESS PARADIGMS

Departing from previous approaches, we impose fairness constraints during the *reward modelling phase*. This guarantees that our reward model is algorithmically fair to provide fair feedback during RL fine-tuning. Let $\mathfrak{I}$ denote a joint distribution over the domain $\mathcal{D}$ of data, where each data sample has the structure $(x, \hat{y}_w, \hat{y}_l, S, U)$. $S \in [p]$ represents a categorical, *sensitive attribute* unfair to use during inference; $U \in [d]$ represents a categorical, *unrestricted attribute* permissible to use. We generalise this formulation to admit multiple sensitive and unrestricted attributes in App. B.2. We define two indicator variables for the preference outcome. The ground-truth human preference is $Y = 1$ for the pair $(\hat{y}_w, \hat{y}_l)$ where humans preferred $\hat{y}_w$ over $\hat{y}_l$. The model's predicted preference is given by $\hat{Y} = \mathbb{1}\{r_\phi(x, \hat{y}_w) > r_\phi(x, \hat{y}_l)\}$. We consider three fair binary classification paradigms depending on the accessibility and modality of attributes:

*(1) Attribute Blind.* $S, U$ are not used for reward assignment. We aim to learn a reward model $r_\phi : \mathcal{X} \times \hat{\mathcal{Y}} \to \mathbb{R}$ that minimises $L(r_\phi; \mathfrak{I})$ and takes $(x, \hat{y})$ as input.

*(2) Attribute Aware.* $S, U$ are accessible and are appended to the input prompt $x$. They are either inferred by an off-the-shelf attributes classifier, or are provided as annotations in the dataset. We aim to learn a reward model $r_\phi : \mathcal{X} \times \hat{\mathcal{Y}} \times [p] \times [d] \to \mathbb{R}$ that minimises $L(r_\phi; \mathfrak{I})$ and takes $(x, \hat{y}, S, U)$ as input.

*(3) Self-critiquing LLMs.* $S, U$ are not provided and must be inferred from input prompt $x$ using an off-the-shelf language model. The natural language descriptions $\hat{S}, \hat{U} \in \mathcal{X}$ of sensitive and unrestricted information (associated with $x$) are appended feature-wise to $x$. We aim to learn a reward model $r_\phi : \mathcal{X} \cdot \hat{S} \cdot \hat{U} \times \hat{\mathcal{Y}} \to \mathbb{R}$ that minimises $L(r_\phi; \mathfrak{I})$ and takes $(x \cdot \hat{s} \cdot \hat{u}, \hat{y})$ as input.

To enforce equity and decrease disparity of inter-group outcomes, we use metrics to characterise the quality of outcomes, such as true positive rate (TPR) (Menon & Williamson, 2018; Agarwal et al., 2018), false positive rate (FPR) (Hardt et al., 2016), and predictive rate (Quadrianto & Sharmanska, 2017). Towards a general framework for fairness statistics, we let values taken by the sensitive attribute $S$ partition the domain $\mathcal{D}$ into $p$ groups $G_i := \{(x, \hat{y}_w, \hat{y}_l, i) \in \mathcal{D}\}$. Then, we follow Celis et al. (2019) to measure $G_i$'s group performance via $q_i^{\mathfrak{I}}(r_\phi) = \mathbb{P}_{\mathfrak{I}}[\mathcal{E} \mid G_i, \mathcal{E}']$ for some events $\mathcal{E}, \mathcal{E}'$, *e.g.* conditioning on unrestricted attributes. Lastly, define the group performance function $q^{\mathfrak{I}} = (q_1^{\mathfrak{I}}(r_\phi), \ldots, q_p^{\mathfrak{I}}(r_\phi))$. We omit $\mathfrak{I}$ when it is contextually clear.

Intuitively, a reward model $r_\phi$ is considered fair with respect to $q$ if $q_i(r_\phi) \approx q_{i'}(r_\phi)$ for all $i, i'$. This indicates that the performance (*e.g.* TPR or FPR) of $r_\phi$ is approximately equal across all subpopulations; the reward model does not overfit the most populous demographic, nor is its performance dependent on specific identities. To measure the fairness of $r_\phi$ for a given group performance function $q$, we follow previous works (Feldman et al., 2015; Menon & Williamson, 2018; Zafar et al., 2017a;b) and consider the $\tau$-rule.

**Definition 2.1.** *(τ-Fair)* A reward model $r_\phi$ achieves $\tau$-fairness w.r.t. $q$ if it satisfies for $\tau \in [0, 1]$,

$$\min_{r_\phi \in \mathcal{R}} L(r_\phi; \mathfrak{I}) \quad \text{s.t.} \quad \max_{i, i' \in [p]} \left| q_i(r_\phi) - q_{i'}(r_\phi) \right| \leq \tau \tag{3}$$

The closer $\tau$ is to 0, the fairer $r_\phi$ is w.r.t $q$; perfect fairness is achieved at $\tau = 0$. Practically, we consider $\tau > 0$ due to known infeasibility, incompatibility and inconsistency issues under perfect fairness (Friedler et al., 2021; Hardt et al., 2016; Kleinberg et al., 2017).

## 2.3 FAIRNESS NOTIONS

We proceed to quantify the fairness violations of reward models, by establishing three notions of group fairness—*demographic parity* (DP), requiring $\hat{Y} \perp\!\!\!\perp S$; *equalised odds* (EO) (Hardt et al., 2016), requiring $\hat{Y} \perp\!\!\!\perp S \mid Y$; *conditional fairness* (CF) (Xu et al., 2020), requiring $\hat{Y} \perp\!\!\!\perp S \mid U$.

**Definition 2.2** (Demographic Parity (DP)). A reward model $r_\phi$ is $\gamma$-DP fair if the group-wise positive rates are nearly equal: $\quad q_i^{\mathrm{dp}}(r_\phi) := \mathbb{P}[\hat{Y} = 1 \mid G_i], \quad \Delta_{\mathrm{dp}}(r_\phi) := \max_{i,i' \in [p]} \left| q_i^{\mathrm{dp}}(r_\phi) - q_{i'}^{\mathrm{dp}}(r_\phi) \right| \leq \gamma.$

**Definition 2.3** (Equalised Odds (EO)). A reward model $r_\phi$ is $\kappa$-EO fair if the TPR/FPR are equalised across groups: $\quad q_{iy}^{\mathrm{eo}}(r_\phi) := \mathbb{P}[\hat{Y} = 1 \mid G_i, Y = y], \quad \Delta_{\mathrm{eo}}(r_\phi) := \max_{i,i' \in [p], y \in \{0,1\}} \left| q_{iy}^{\mathrm{eo}}(r_\phi) - q_{i'y}^{\mathrm{eo}}(r_\phi) \right| \leq \kappa.$

**Definition 2.4** (Counterfactual Fairness (CF)). A reward model $r_\phi$ is $\mu$-CF fair (conditional on $U$) if, for each $j \in [d]$, the group-conditioned positive rates match across groups:

$$q_{ij}^{\mathrm{cf}}(r_\phi) := \mathbb{P}[\hat{Y} = 1 \mid G_i, U = j], \quad \Delta_{\mathrm{cf}}(r_\phi) := \max_{i,i' \in [p], j \in [d]} \left| q_{ij}^{\mathrm{cf}}(r_\phi) - q_{i'j}^{\mathrm{cf}}(r_\phi) \right| \leq \mu.$$

We also use an averaged version: $\quad \Delta_{\mathrm{cf}}^{\mathrm{avg}}(r_\phi) := \mathbb{E}_U \left[ \max_{i,i' \in [p]} \left| q_{iU}^{\mathrm{cf}}(r_\phi) - q_{i'U}^{\mathrm{cf}}(r_\phi) \right| \right] \leq \mu.$

## 3 FAIRNESS-AWARE REWARD OPTIMIZATION

A well-aligned reward model must capture multifaceted features of human preferences: *(i) ordinal*, correctly ranking preferred responses; *(ii) cardinal*, accurately modeling the margin of these preferences; and *(iii) fair*, ensuring that accuracy is consistent across demographics. Existing methods often focus only on ordinal accuracy, leading to poorly calibrated or systematically biased models. We proceed to develop `FARO`: enforcing fairness in the LLM by guaranteeing algorithmic fairness in the reward function. We augment standard preference learning with fairness constraints, reformulating this as a Lagrangian minimax problem:

$$\min_\phi \max_{\lambda \geq 0} L_{\mathrm{NLL}}(\phi) + \lambda^\top C_{\mathrm{fairness}}(\phi) \tag{4}$$

Here, we optimise over the model parameters $\phi$, which define a reward model $r_\phi(x, y)$ assigning a scalar score to a response. This reward model induces a probabilistic preference model $p_\phi(\hat{y}_w \succ \hat{y}_l \mid x)$, typically via the Bradley-Terry model (Eq. 1). Both terms in the Lagrangian depend on these preference probabilities: $L_{\mathrm{NLL}}(\phi)$ is the negative log-likelihood of $p_\phi$ with respect to human preference data, and $C_{\mathrm{fairness}}(\phi)$ is a vector of fairness constraint violations, computed as expectations of $p_\phi$ across demographic groups. The dual variables $\lambda$ are learned penalties applied to these violations.

This optimization has two challenges: *(1) non-differentiable constraints, (2) quadratic complexity*. Fairness constraints are often defined on empirical classification rates, which have zero gradients almost everywhere and are unsuitable for optimisation. We instead use a differentiable proxy for these rates, defined by the model's expected preference probability, $\mathbb{E}[p_\phi(\hat{y}_w \succ \hat{y}_l \mid x)]$. Moreover, to avoid quadratic $O(p^2)$ complexity from all pairwise group comparisons, we employ the anchoring trick (Jagielski et al., 2019) and reduce the number of constraints to $O(p)$ without loss of generality for feasibility. If constraints $|q_1(r_\phi) - q_i(r_\phi)| \leq \gamma_i$ hold for all $\geq 2$ hold, then by the triangle inequality any two groups $i, j \geq 2$ satisfy $|q_i(r_\phi) - q_j(r_\phi)| \leq \gamma_i + \gamma_j$.

The final `FARO` objective incorporates these solutions. Given a set of non-uniform fairness tolerances $\gamma_i, \kappa_i, \mu_{ij}$, the fairness constraint vector $C_{\mathrm{fairness}}(\phi)$ is defined for one of the following standards:

*(1) DP:* The vector $C_{\mathrm{dp}}(\phi)$ contains the $2(p-1)$ constraints derived from the inequalities: $|q_1^{\mathrm{dp}}(r_\phi) - q_i^{\mathrm{dp}}(r_\phi)| \leq \gamma_i \quad$ for $i \in \{2, \ldots, p\}$.

*(2) EO:* The vector $C_{\mathrm{eo}}(\phi)$ is defined analogously, with expectations taken conditioned on the human preference label $Y = y$: $|q_1^{\mathrm{eo}}(r_\phi \mid Y = y) - q_i^{\mathrm{eo}}(r_\phi \mid Y = y)| \leq \kappa_i \quad$ for $i \in \{2, \ldots, p\}$, $y \in \{0, 1\}$.

*(3) CF:* The vector $C_{\mathrm{cf}}(\phi)$ is defined by conditioning on an unrestricted attribute $U = j$: $|q_{1j}^{\mathrm{cf}}(r_\phi \mid U = j) - q_{ij}^{\mathrm{cf}}(r_\phi \mid U = j)| \leq \mu_{ij} \quad$ for $i \in \{2, \ldots, p\}$, $j \in [d]$.

## 4 THEORETICAL ANALYSIS

To solve problem 4, we adapt the proxy-Lagrangian gradient descent–ascent (ProxyGDA) method by Cotter et al. (2019a;b). Specifically, we instantiate the two-player game in Eq. 4 with FARO's fairness constraints and analyse the regret bounds of the resulting dynamics. While the algorithmic template is standard, its application to fair reward modelling and downstream RLHF is novel to our work. We establish four guarantees: (i) the FARO-learned reward satisfies DP/EO/CF constraints up to a controllable, diminishing slack; (ii) RL fine-tuning with a KL penalty induces an accuracy-fairness trade-off; (iii) using a FARO-fair reward improves downstream policy fairness compared to an unconstrained reward; and (iv) varying tolerance and regularisation parameters in FARO traces a non-empty Pareto frontier of optimal solutions.

---

**Algorithm 1** PROXYGDA FOR FARO ($R \in \mathbb{R}_+$, $L_{\mathrm{faro}} : \Phi \times \Lambda \to \mathbb{R}$, $T \in \mathbb{N}$, $\eta_\lambda, \eta_\phi, \varepsilon_{\mathrm{rel}} \in \mathbb{R}_+$)

---

1: Initialise $\lambda^{(1)} = 0$
2: **for** $t \in [T]$ **do**
3:     Initialise $\phi^{(t,0)}$ randomly
4:     **repeat**
5:         $\phi^{(t,k+1)} = \phi^{(t,k)} - \eta_\phi \nabla_\phi L_{\mathrm{faro}}(\phi^{(t,k)}, \lambda^{(t)})$
6:     **until** $\frac{\left| L_{\mathrm{faro}}(\phi^{(t,k)}, \lambda^{(t)}) - L_{\mathrm{faro}}(\phi^{(t,k-1)}, \lambda^{(t)}) \right|}{\max\{1, |L_{\mathrm{faro}}(\phi^{(t,k)}, \lambda^{(t)})|\}} \leq \varepsilon_{\mathrm{rel}}$
7:     Let $\phi^{(t)} = \phi^{(t,k)}$
8:     Update $\lambda^{(t+1)} = \Pi_\Lambda \left( \lambda^{(t)} + \eta_\lambda \nabla_\lambda L_{\mathrm{faro}}(\phi^{(t)}, \lambda^{(t)}) \right)$    ▷ Projection onto $\Lambda = \{\lambda \geq 0 : \|\lambda\|_\infty \leq R\}$
9: **end for**
10: **return** averaged iterate $\bar{\phi} = \frac{1}{T} \sum_{t=1}^{T} \phi^{(t)}$ (or best iterate by validation)

---

### 4.1 REWARD-LEVEL FAIRNESS CERTIFICATES

Algorithm 1 describes the gradient descent–ascent method for optimizing the FARO Lagrangian, $L_{\mathrm{faro}}$. The inner loop (Lines 4–6) iteratively finds an approximate minimiser of the loss with respect to the model parameters $\phi$, stopping when a relative tolerance $\varepsilon_{\mathrm{rel}}$ is met. This process yields a *$\rho$-approximate solution* $\phi^{(t)}$, where the absolute approximation error $\rho$ is implicitly controlled by $\varepsilon_{\mathrm{rel}}$. Based on this procedure, we can certify the fairness of the resulting reward model.

**Proposition 4.1** (Population fairness certificate for FARO). *Let $\bar{\phi}$ be the averaged iterate returned by* PROXYGDA. *Then with probability at least $1 - \delta$, the population fairness violations of $r_{\bar{\phi}}$ satisfy*

$$\max_c \Delta^c(\bar{\phi}) \ \leq \ \rho + \widetilde{O}\left(\frac{R}{\sqrt{T}}\right) + O\left(\sqrt{\frac{\log(1/\delta)}{n_{\min}}}\right),$$

*where $c \in \{\mathrm{dp}, \mathrm{eo}, \mathrm{cf}\}$ and $n_{\min} = \min_i n_i$ is the sample size of the smallest sensitive subgroup. Thus $r_{\bar{\phi}}$ is $\gamma$-DP / $\kappa$-EO / $\mu$-CF fair up to a controllable slack consisting of the inner-loop optimisation error $\rho$, convergence error, and a generalisation gap that vanishes with more data (see App. C.1).*

**Corollary 4.2** (Group-wise $\tau$-rules). *For any feasible solution to the group-fair DP program in Eq. 4 with ordered allowances $\{\gamma_i\}$, the learned reward $r_\phi$ satisfies, with slack $\varepsilon_T = \rho + O(RG\sqrt{k/T})$ (Prop. 4.1),*

$$|q_i^{\mathrm{dp}}(r_\phi) - q_j^{\mathrm{dp}}(r_\phi)| \leq \gamma_i + \gamma_j + 2\varepsilon_T, \qquad\qquad \forall i, j \geq 1,$$
$$q_1^{\mathrm{dp}}(r_\phi) - (\gamma_i + \varepsilon_T) \leq q_i^{\mathrm{dp}}(r_\phi) \leq q_1^{\mathrm{dp}}(r_\phi) + (\gamma_i + \varepsilon_T), \qquad\qquad \forall i \geq 2.$$

Together, Prop. 4.1 and Cor. 4.2 certify that FARO yields a reward model that is DP/EO/CF-fair up to a controllable slack $\varepsilon_T = \rho + O(RG\sqrt{k/T})$, plus a statistical term $O(\sqrt{\frac{\log(1/\delta)}{n_{\min}}})$ for population guarantees (App. C.1- C.2). The slack is governed by the optimisation budget $T$, inner-loop tolerance $\rho$, and data balance.

## 4.2 FINETUNING INDUCES AN ACCURACY–FAIRNESS TRADE-OFF

Having engineered a fair reward model $r_\phi$, we now analyse how it can be used to induce fairness in a performant but potentially biased LLM policy, $\pi_{\text{ref}}$. The process of RL fine-tuning creates three-way tension: *alignment*, *i.e.* maximising the score from the fair reward model $r_\phi$; *performance retention*, *i.e.* staying close to the strong reference policy $\pi_{\text{ref}}$ that captures general capabilities; and *final policy fairness*, *i.e.* ensuring that the resulting fine-tuned policy $\pi_\beta$ is itself fair. The KL-regularised objective from Eq. 2 illustrates this trade-off. The fair reward $r_\phi$ pulls the policy towards a fair region, while the KL term acts as an anchor to the performant $\pi_{\text{ref}}$, controlled by the hyperparameter $\beta$. A small $\beta$ allows greater deviation towards the fair reward (potentially improving fairness and accuracy), whereas a large $\beta$ keeps the policy close to $\pi_{\text{ref}}$ (preserving task behaviour but also its unfairness).

We measure divergence between $\pi, \pi'$ by the policy-induced KL $D_{\text{KL}}(\pi \| \pi') = \mathbb{E}_{x \sim D}[D_{\text{KL}}(\pi(\cdot \mid x) \| \pi'(\cdot \mid x))]$. By Pinsker's inequality (Lemma C.1), deviations from $\pi_{\text{ref}}$ bound changes in group-level probabilities.

**Proposition 4.3** (KL-regularised trade-off and fairness drift). *Let $\pi_\beta$ be any maximizer of the KL-regularized objective $\mathcal{J}_\beta(\pi) = \mathbb{E}_{x,a \sim \pi}[r_\phi(x,a)] - \beta \, D_{\text{KL}}(\pi \| \pi_{\text{ref}})$. Then:*

1. *(Monotonicity) If $\beta_1 > \beta_2 > 0$ then $D_{\text{KL}}(\pi_{\beta_1} \| \pi_{\text{ref}}) \leq D_{\text{KL}}(\pi_{\beta_2} \| \pi_{\text{ref}})$.*

2. *(Fairness drift) The fairness violation of the final policy $\pi_\beta$ is bounded by the violation of the initial reference policy, $\Delta(\pi_{\text{ref}})$, plus a drift term: $\Delta(\pi_\beta) \leq \Delta(\pi_{\text{ref}}) + \sqrt{2 \, D_{\text{KL}}(\pi_\beta \| \pi_{\text{ref}})}$.*

Prop. 4.3 reveals the dual role of the KL term: beyond regularising the policy update to preserve the capabilities of $\pi_{\text{ref}}$, it provides a worst-case guarantee that the fairness violation will not degrade arbitrarily. The trade-off for a practitioner is thus in the choice of $\beta$, which balances the pursuit of higher reward (permitting a larger KL divergence) against maintaining a tighter fairness bound. This reframes the objective from preserving the biased reference policy to controlling the magnitude of the departure from it. This raises the crucial question of how to ensure this "drift" is a beneficial move towards fairness. We address this in Thm. 4.4, which shows that updates guided by a FARO-fair reward model provably improve downstream policy fairness.

## 4.3 FAIRER RL POLICIES WHEN USING FAIR REWARDS

We have established that RL fine-tuning involves a controlled "drift" away from a performant but potentially biased reference policy, $\pi_{\text{ref}}$. Prop. 4.3 showed that the magnitude of this drift, controlled by $\beta$, has a bounded effect on the final policy's fairness. This raises the central question: if we guide this drift with a FARO-fair reward model, does it actually produce a fairer final LLM?

We answer in the affirmative, showing that the fairness engineered into the reward model provably transfers to the fine-tuned policy. This is a critical result, as it guarantees that our efforts at the reward-modeling stage are not "lost in translation" during the complex dynamics of RL optimisation. It shows that using a fair reward is demonstrably better than using an unconstrained one.

**Theorem 4.4** (Reward-to-Policy Fairness Transfer). *Let $r_{plain}$ be a reward model trained to optimise only the preference loss (Eq. 1) on a given dataset, and let $r_\phi$ be the FARO-fair reward model trained on the same data with an additional fairness constraint. Let their resulting fairness violations be $\Delta(\pi_{\beta_{plain}})$ and $\Delta(\pi_{\beta_{fair}})$ respectively, after fine-tuning from the same $\pi_{ref}$ with the same KL-penalty $\beta$. Under standard monotonicity assumptions, the violations are related by $\Delta(\pi_\beta^{\text{fair}}) \leq \Delta(\pi_\beta^{\text{plain}}) + \varepsilon_T$.*

$\varepsilon_T$ is the fairness violation slack of the reward model $r_\phi$ from Prop. 4.1. For any given level of fine-tuning (*i.e.* for any fixed $\beta$), replacing a standard, unconstrained reward with a FARO-fair reward will improve (at worst, not harm) the downstream fairness of the resulting LLM policy, up to the small slack $\varepsilon_T$. A fair reward function makes the final policy fairer. The guarantees we establish at the reward level propagate through the RL fine-tuning process, providing a principled mechanism for producing fairer policies in practice (App. C.4).

### 4.4 EXISTENCE OF PARETO-OPTIMAL OPERATING POINTS

Consider the bi-objective problem of minimising (error, fairness), where "error" is a suitable accuracy metric and "fairness" is one of $\Delta_{dp}, \Delta_{eo}, \Delta_{cf}$. We establish the existence of a non-empty Pareto frontier, ensuring that there are well-defined operating points trading off fairness and accuracy.

**Proposition 4.5** (Non-empty Pareto frontier). *Varying FARO's fairness tolerance schedules $\{\gamma_i\}, \{\kappa_i\}, \{\mu_{ij}\}$ and the KL-regularisation parameter $\beta$ within compact sets traces a non-empty, continuous Pareto frontier in the (error, fairness) objective space.*

This guarantee arises from a standard topological argument (full proof in App C.5). The space of hyperparameters is compact by definition. The mapping from these parameters to the resulting optimal policy, $\pi^*$, is continuous by Berge's Maximum Theorem, as is the subsequent mapping from the policy to its (error, fairness) evaluation. The continuous image of a compact set is also compact; therefore, the set of all achievable outcomes is a compact set in $\mathbb{R}^2$, which ensures the existence of a non-empty Pareto frontier.

As tolerances $(\gamma, \kappa, \mu) \to 0$ and $\beta \to \infty$, the policy remains close to the unfair reference $\pi_{ref}$. Conversely, as $\beta \to 0$, the policy utilises the fair reward $r_\phi$, improving fairness with controlled deviation from $\pi_{ref}$. FARO efficiently traverses the continuous trade-off space, and yields Pareto-efficient policies in RL finetuning.

## 5 EXPERIMENTS

We evaluate each setting FARO on safety oriented datasets. For each run we optimize a single fairness family FARO_dp, FARO_eo, or FARO_cf. We finetune an instruction tuned language model with reward modeling and use the learned reward to assess multiple choice selection or to rerank sampled generations at inference.

**Finetuning Dataset.**  We train the reward model on PRISM (Kirk et al., 2024), a pairwise preference corpus that is grouped by sociodemographic attributes. Our implementation follows the proxy Lagrangian with anchoring. For a chosen family we form anchored constraints over the differentiable preference probability, learn nonnegative dual variables with projection, and optimize the Bradley Terry negative log likelihood plus the active constraint term. Training uses a value head on top of the policy.

**Evaluation Datasets.**  We evaluate with dataset specific quality and fairness metrics. On *BBQ* (Parrish et al., 2022), we use the Ambiguous and Disambiguated settings. We report top-1 accuracy and the official bias scores for both settings. For BBQ we score multiple choice options by the reward and select the argmax.

**Models.**  We use three public instruction tuned models: Gemma-2-2B (Team et al., 2024), Phi-3-Mini (Abdin et al., 2024), and Qwen-2.5-1.5B (Team, 2024). Reward modeling is performed with a causal language model and value head. For evaluation we either rank options by the reward, or generate with the policy and optionally apply reward based reranking as above. We include a baseline comparison with the original language model: for Gemma, we use reported scores for the BBQ evaluation; for Phi-3 and Qwen-2.5, we extract findings using the same approaches in Parrish et al. (2022) and report scores in the table independently.

We show results in Table 2. We find that FARO allows us to consistently reduce bias as given by the bias score while preserving general accuracy.

### 5.1 OPTIMISING THE ERROR VS. FAIRNESS-VIOLATION TRADE-OFF

We show a comparison between our top-1 accuracy and DP loss for Gemma in Figure 2. This is varied across several $\beta$ values to understand how $\beta$ affects the relationship between fairness and accuracy for Gemma. We evaluate the fairness using DP and evaluate the accuracy using the accuracy of BBQ for Gemma. We include

Table 2: **FARO optimises fairness while preserving performance**. We measure fairness on BBQ after reward-optimising on PRISM. We find that FARO allows us to significantly improve over the base model with regards to bias scores. This also seems to correlate with changes in the scores of DP, EO, and CF.

| Model | Disamb Top-1 ($\uparrow$) | Ambig Top-1 ($\uparrow$) | Ambig Bias Score ($\downarrow$) | Disambig Bias Score ($\downarrow$) | $\Delta_{\text{DP/EO/CF}}$ |
|---|---|---|---|---|---|
| Gemma-2-2b-it | 83.20 | 63.91 | 14.73 | -0.811 | N/A |
| Gemma-2-2b-it – FARO (DP) | **83.93** | 63.20 | **6.81** | **-1.01** | 0.55 |
| Gemma-2-2b-it – FARO (CF) | 83.10 | 62.86 | 10.55 | -0.965 | 0.41 |
| Gemma-2-2b-it – FARO (EO) | 82.71 | **63.72** | 12.96 | -0.822 | 0.44 |
| Phi-3-Mini | 71.92 | 42.14 | 11.91 | 1.42 | N/A |
| Phi-3-Mini – FARO (DP) | **71.99** | **46.55** | 9.15 | 1.01 | 0.21 |
| Phi-3-Mini – FARO (EO) | 70.05 | 44.01 | 10.86 | 1.04 | 0.18 |
| Phi-3-Mini – FARO (CF) | 71.73 | 45.92 | **9.01** | **0.93** | 0.37 |
| Qwen-2.5-1.5B | 74.14 | 58.97 | 11.44 | -.0922 | N/A |
| Qwen-2.5-1.5B– FARO (DP) | **75.11** | **59.18** | 9.11 | -.104 | 0.26 |
| Qwen-2.5-1.5B – FARO (EO) | 74.06 | 57.66 | 10.87 | -0.100 | 0.054 |
| Qwen-2.5-1.5B – FARO (CF) | 73.12 | 58.91 | **8.04** | **-0.155** | 0.091 |

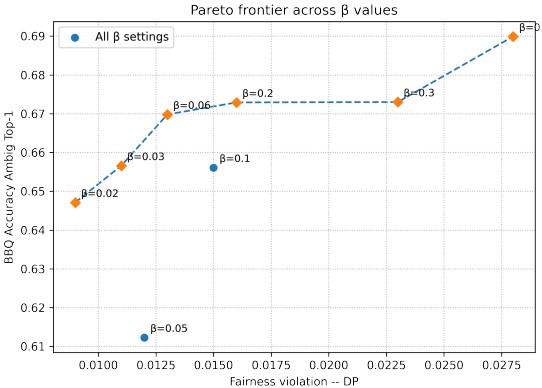

Figure 2: **Pareto Frontier of fairness and accuracy.** We vary $\beta$ and use FARO_DP as the reward for Gemma on PRISM. We plot the fairness violation and BBQ Top-1 accuracy for the ambiguous dataset, and compute the pareto optimal set of $\beta$s by finding all dominated points where all neighboring points are strictly better.

the pareto frontier by finding all points which are dominated by other points around them and removing them. Surprisingly, we see some sensitivity to $\beta$ e.g. with 0.05. This could be due to some tuning whereby smaller $\beta$ is not well balanced between the fairness and accuracy considerations.

## 6 CONCLUSION

We tackle the issue of demographic bias in LLM alignment, which propagates from skewed or prejudiced human preference data. We argue that existing interventions are unable to address all axes of the problem, where a suitable reward model must be simultaneously *ordinal, cardinal, and fair*. Towards codifying and reinforcing fair behaviour, we introduce FARO, an in-processing framework that directly embeds *algorithmic fairness constraints into the reward modeling objective*. Our theoretical analysis provides several guarantees; notably, that the fairness engineered into the reward model provably *transfers to the fine-tuned policy*, and that *a Pareto frontier of optimal solutions exists*. We validate this theory across the BBQ benchmark and three LLMs, confirming that FARO significantly reduces biased or prejudiced generations whilst preserving model quality. This work offers a principled and verifiable path toward more equitable LLMs that are fair by design.

## ETHICS STATEMENT

This paper investigates fairness shortcomings in LLM alignment and proposes improvements via FARO, an in-processing intervention with fairness constraints during RLHF's reward modelling phase. We contribute 3 desirable properties—the ability to correct for human bias in datasets; to conduct fairness-aware optimisation in an annotation efficient manner; to derive reward models that are algorithmically fair with high Pareto-efficiency. While mathematical guarantees can guard against worst-case examples of egregious discrimination, fairness is an inherently societal concept; theoretical models must be continuously updated by inter-disciplinary research. Algorithms like FARO should be used to complement – not replace – other fairness guardrails (*e.g.* data-filtering, unlearning, calibration). A fair model can still be misused; due diligence, rigorous auditing, collecting and incorporating user feedback are as important as ever before.

## REPRODUCIBILITY STATEMENT

We report experimental details on the considered datasets, models and baselines, and hyperparameters for FARO in Section 5 and App. D.1. We discuss (algorithmic) metrics of fairness in Section 2.2; we further discuss the limitations of measures of fairness in Section 6. Our theoretical arguments are substantiated by quantitative and qualitative results, with full proofs provided in App. C. Upon acceptance, we will open source the FARO framework code and FARO-trained reward models for scientific collaboration.

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

## A  EXTENDED RELATED WORK

**Why Preference Alignment is Not "Fair Enough".**   Using preference datasets for fairness-alignment is vulnerable to *selection bias* from data collection oversight; *popularity bias* from disproportional survey participation; *cognitive bias* from prejudiced human annotators[1]. Flawed data collection induces selection bias, where response data is biased by preferences of the surveyed demographic and encodes spurious correlations (Ovaisi et al., 2020; Wang et al., 2021; Liu et al., 2022); skewed survey strategies lead to popularity bias, where preference data is sparse, long-tailed and lacks coverage for less-preferred or uncommon responses, leading to unpredictable behaviour under distribution shifts or edge cases (Chen et al., 2023; Zhao et al., 2023; Naghiaei et al., 2022). Beyond statistical dataset biases, previous works also deal with a lack of fairness in LLMs' judgement calls, as a product of defective training and alignment procedures. Surveys on fairness (Mehrabi et al., 2021; Gallegos et al., 2024) reveal cognitive biases in LLM judges that mirror human prejudice, leading

---

[1]We refer readers to Gallegos et al. (2024) for a thorough and insightful breakdown of the metrics, datasets, mitigation techniques and open problems concerning LLM bias and fairness.

to disparate treatment on unfair bases of gender, authority, beauty standards, misinformation (Chen et al., 2024a; Koo et al., 2024; Zheng et al., 2023). Since real-world datasets are rampant with discriminatory examples, bias arising from these skewed representations are encoded into the reward model (Wang et al., 2023b; Liu et al., 2020); the reward model propagates undesirable biases to the LLM through RL fine-tuning and reinforces unfair behaviour (Blodgett et al., 2020; 2021). Without an internal constitution, our LLMs are being corrupted by—instead of correcting—instances of bias, unfairness and prejudice in big data. Enter the role of theory as a tool to specify, certify and codify principles of equality into the LLM constitution.

**Algorithmic Fairness.** Fairness concerns equalising the treatment and consideration of people, identified by their (protected) demographic attributes, such as gender, race, age (Commission, 1964). Fairness-aware algorithms intervene on the learning problem to avoid *disparate treatment* of people, with hopes of also reducing *disparate impact* of decision-making outcomes (Barocas & Selbst, 2016). Notable definitions of group-fairness include demographic parity (Dwork et al., 2012), equalised odds and equal opportunity (Hardt et al., 2016), calibration (Kleinberg et al., 2017); different approaches to fairness are benchmarked on real-world datasets, including TransUnion TransRisk (Avery et al., 2009), UCI Adult (Kohavi & Becker, 1996), Dutch census (Center, 2019), COMPAS (Larson et al., 2016), ACS PUMS (Ding et al., 2021). Towards mitigating group-wise disparate treatment, there are three types of fairness interventions—*pre-processing, in-processing* and *post-processing* (Zeng et al., 2022; Pleiss et al., 2017). This mirrors the dilemma of fairness-aware processing in LLMs, where the timing of when to intervene has crucial impacts on both the algorithm (computational and sample efficiency, optimisation stability) and its outcomes (whether it has theoretical guarantees, its Pareto-efficiency). Pre-processing aims to filter away latent biases in training data through transformations (Feldman et al., 2015; Lum & Johndrow, 2016; Johndrow & Lum, 2019; Calmon et al., 2017), fair representations (Zemel et al., 2013; Louizos et al., 2015; Creager et al., 2019) and fair generative modelling (Xu et al., 2018; Sattigeri et al., 2019; Jang et al., 2021). Although such methods are broadly applicable to any learning problem, pre-processing requires an extra, expensive pass over data, and lacks formal guarantees of fairness, since disparities may persist even post-filtering (Locatello et al., 2019). A different approach is to post-process a trained model by shifting its decision boundary – particularly with group-wise thresholding rules (Fish et al., 2016; Corbett-Davies et al., 2017; Menon & Williamson, 2018; Chzhen et al., 2019; Jang et al., 2022) – to adjust for fairness. However, previous work has demonstrated that post-processing is unable to achieve optimality in both error-calibration and fairness; that post-processing for one particular notion of fairness could be in contradiction other important but incompatible notions (Chouldechova, 2017; Kleinberg et al., 2017; Woodworth et al., 2017; Corbett-Davies et al., 2017).

**Self-critiquing LLMs.** One exciting direction concerns LLMs self-improvement and self-correction by critiquing their own outputs. Self-criticism generates new instructions and the model is realigned to the new instructions (Zheng et al., 2023; Wang et al., 2023c; Honovich et al., 2023). In fairness-aligned optimisation, this involves using context-dependent techniques (Wang et al., 2024) to first, infer sensitive and unrestricted attributes from input prompts; then, recast reward modelling as an "Attribute Aware", in-processing problem; finally, iteratively self-assess its Pareto-efficiency and adjust the attributes-classifier to issue systematic updates to the self-rewarding mechanism (Yuan et al., 2024). Though it is yet unclear whether LLMs as self-critics can be immune from evaluation bias, human/dataset bias and positional instability issues (Wang et al., 2023a; Koo et al., 2024; Chen et al., 2024b; Sun et al., 2024), we are optimistic that a hybrid approach structured with algorithmic fairness could reveal new strategies for robust, stable and fair self-alignment.

# B  DERIVATIONS

## B.1  FARO FOR DIRECT PREFERENCE OPTIMISATION

DPO-like frameworks reframe the RL problem by instead expressing the reward model in terms of the reference and optimal policies. The derivation begins by noting that the optimal policy for the KL-constrained

reward maximisation objective (Eq. 2) is a Gibbs distribution. This allows the reward difference between winning and losing responses to be defined purely by the policies themselves. By substituting this policy-based reward expression into the BT-model, DPO arrives at a simple negative log-likelihood loss that is optimised directly with respect to the policy's parameters:

$$L_{\text{DPO}}(\pi_\theta; \pi_{\text{ref}}) = -\mathbb{E}_{(x, y_w, y_l) \sim D} \left[ \log \sigma \left( \beta \log \frac{\pi_\theta(y_w \mid x)}{\pi_{\text{ref}}(y_w \mid x)} - \beta \log \frac{\pi_\theta(y_l \mid x)}{\pi_{\text{ref}}(y_l \mid x)} \right) \right]. \quad (5)$$

For frameworks such as DPO , KTO and GRPO, we may directly combine their standard loss functions with a fairness penalty term, where the implicit reward is substituted into the fairness proxy:

$$L_{\text{FARO-\{DPO, KTO, GRPO\}}}(\pi_\theta, \lambda) = L_{\{\text{DPO, KTO, GRPO}\}}(\pi_\theta) + \lambda^\top C_{\text{fairness}}(\pi_\theta) \quad (6)$$

**FARO-DPO.** In DPO (Rafailov et al., 2023), the fairness violation vector, $C_{\text{fairness}}(\pi_\theta)$, is composed of constraints based on the policy-dependent fairness proxy, $q_i(\pi_\theta)$:

$$q_i(\pi_\theta) := \mathbb{E}_{(x, y_w, y_l) \sim D_i} \left[ \sigma \left( \beta \log \frac{\pi_{\text{ref}}(y_w \mid x)}{\pi_\theta(y_w \mid x)} - \beta \log \frac{\pi_{\text{ref}}(y_l \mid x)}{\pi_\theta(y_l \mid x)} \right) \right] \quad (7)$$

**FARO-KTO.** KTO (Ethayarajh et al., 2024) uses single responses – as opposed to pairs – labelled as desirable or undesirable. FARO can be applied by constraining the average reward for desirable (or undesirable) examples to be equal across groups. The fairness constraint is defined on the implicit KTO reward, $r_{\text{KTO}}(x, y) = \beta \log \left( \frac{\pi_\theta(y \mid x)}{\pi_{\text{ref}}(y \mid x)} \right)$. For a desirable example ($Y = 1$), the fairness proxy for group $G_i$ is:

$$q_i(\pi_\theta) := \mathbb{E}_{(x, y) \sim D_i, Y=1} \left[ \sigma \left( \beta \log \frac{\pi_{\text{ref}}(y \mid x)}{\pi_\theta(y \mid x)} \right) \right] \quad (8)$$

**FARO-GRPO.** GRPO (Shao et al., 2024) is designed for group-wise preference data. FARO extends this by ensuring that within each group, the preference margins are consistent with the global fairness standard. The fairness constraints can be defined similarly to FARO-DPO but the expectations for the proxy $q_i(\pi_\theta)$ are taken over the specific preference distributions for each group $G_i$.

## B.2 GENERALISING TO MULTIPLE ATTRIBUTES.

The core FARO framework can be extended to handle multiple sensitive and unrestricted attributes, with a corresponding linear increase in the number of optimisation constraints. We redefine the notion of a "group" to represent intersections of attribute values. We assume a setup of $N$ sensitive attributes, $S_1, S_2, \ldots, S_N$, where each attribute $S_n$ can take one of $p_n$ categorical values. Similarly, we assume $K$ unrestricted attributes, $U_1, U_2, \ldots, U_K$, where each attribute $U_k$ can take one of $d_k$ values. A data sample now has the structure $(x, \hat{y}_w, \hat{y}_l, S_1, \ldots, S_N, U_1, \ldots, U_K)$.

Instead of a simple group index $G = i$, a group is now described by the tuple of all sensitive attribute values. A specific intersectional group $G_i$ corresponds to a particular combination of values $(s_1, s_2, \ldots, s_N)$, where $s_n$ is a value for attribute $S_N$. The total number of sensitive groups becomes the product of the number of categories for each sensitive attribute: $p = p_1 \times p_2 \times \cdots \times p_N$. For instance, if we have two sensitive attributes Gender ($S_1 \in$ {Male, Female, Non-binary}) and Employment status ($S_2 \in$ {Employee, Self-employed, Not employed}), the total number of intersectional groups is $3 \times 3 = 9$. A group $G_i$ would, for instance, be "male employee" or "self-employed female".

Fairness constraints are applied over this set of $p$ intersectional groups with the same anchoring technique [2]:

---

[2] We select one attribute combination as the reference (*e.g.* "Male employee") and constrain other groups relative to it.

- *Demographic Parity (DP).* Constraints are applied to the $p$ intersectional groups, incurring $2(p-1)$ constraints for each of the non-reference groups:

$$\left| q_{\text{ref}}^{\text{dp}}(r_\phi) - q_i^{\text{dp}}(r_\phi) \right| \leq \gamma_i$$

- *Equalised Odds (EO).* Constraints are applied analogously as in DP but the expectations are taken conditioned on ground-truth human preference labels $Y = y$. This serves to equalise the model's TPR and FPR across groups, incurring $2 \cdot 2(p-1)$ constraints from the inequalities:

$$\left| q_1^{\text{eo}}(r_\phi \mid Y = y) - q_i^{\text{eo}}(r_\phi \mid Y = y) \right| \leq \kappa i$$

- *Conditional Fairness (CF).* Constraints are applied for each sensitive group, conditioned on each combination tuple of unrestricted attributes, $(u_1, u_2, \ldots, u_K)$; this incurs $2d(p-1)$ constraints:

$$\left| q_{\text{ref},j}^{\text{cf}}(r_\phi) - q_{i,j}^{\text{cf}}(r_\phi) \right| \leq \mu_{ij}$$

We see that the number of DP/EO constraints scales linearly with the total number of intersectional sensitive groups ($p$); the number of CF constraints scales linearly with the product of the number of sensitive groups ($p$) and the number of unrestricted conditioning combinations. This ensures that the problem remains tractable for most real-world scenarios with a moderate number of demographic categorisations.

## C  PROOFS

### C.1  PROOF OF PROPOSITION 4.1 (POPULATION FAIRNESS CERTIFICATE FOR FARO)

*Proof.* The FARO objective can be written as the minimax problem $\min_\phi \max_{\lambda \in \Lambda} \left( - \mathbb{E}[r_\phi] + \lambda^\top \big( \mathbf{q}(\phi) - \gamma \big) \right)$, where $\Lambda = \{\lambda \in \mathbb{R}^k : 0 \leq \lambda_j \leq R\}$. ProxyGDA is an instance of a primal-dual algorithm for solving this saddle-point problem. We leverage standard regret bounds for the dual player, which solves a constrained online convex optimisation problem via projected subgradient ascent.

Let $\phi^{(t)}$ be a $\rho$-approximate primal solution at step $t$ for a given $\lambda^{(t)}$. The dual player performs the update $\lambda^{(t+1)} = \Pi_\Lambda \big[ \lambda^{(t)} + \eta_\lambda \big( \mathbf{q}(\phi^{(t)}) - \gamma \big) \big]$, where the subgradient is $g^{(t)} = \mathbf{q}(\phi^{(t)}) - \gamma$. Assuming bounded gradients $\|g^{(t)}\|_2 \leq G$, standard regret analysis for online projected gradient ascent (e.g., Cotter et al. 2019a) implies that for any $\lambda^* \in \Lambda$:

$$\sum_{t=1}^{T} (\lambda^{(t)})^\top g^{(t)} \geq \sum_{t=1}^{T} (\lambda^*)^\top g^{(t)} - \frac{\|\lambda^{(1)} - \lambda^*\|_2^2}{2\eta_\lambda} - \frac{\eta_\lambda}{2} \sum_{t=1}^{T} \|g^{(t)}\|_2^2. \tag{9}$$

Choosing $\lambda^* = 0$ and $\lambda^{(1)} = 0$, and noting that $\sum_{t=1}^{T} \|g^{(t)}\|_2^2 \leq TG^2$, we obtain

$$\sum_{t=1}^{T} (\lambda^{(t)})^\top g^{(t)} \geq -\frac{\eta_\lambda}{2} TG^2.$$

By convexity of $\mathbf{q}(\cdot)$ and Jensen's inequality, the violation at the averaged $\bar{\phi} = \frac{1}{T} \sum_{t=1}^{T} \phi^{(t)}$ is bounded by

$$\mathbf{q}(\bar{\phi}) - \gamma \leq \frac{\text{diam}(\Lambda)^2}{2\eta_\lambda T} + \frac{\eta_\lambda G^2}{2}. \tag{10}$$

Since $\text{diam}(\Lambda)^2 \leq kR^2$, setting $\eta_\lambda = \frac{R\sqrt{k}}{G\sqrt{T}}$ balances the terms, yielding

$$\mathbf{q}(\bar{\phi}) - \gamma \leq \frac{RG\sqrt{k}}{\sqrt{T}}.$$

Adding the $\rho$-error from approximate primal solves, the total violation for any proxy constraint is

$$\varepsilon_T = \rho + O\left(\frac{RG\sqrt{k}}{\sqrt{T}}\right).$$

Thus, as $T \to \infty$, the proxy violation converges to $\rho$. $\qquad\square$

**Clarification (proxies vs. true constraints).** The above analysis certifies feasibility with respect to the *proxy constraints* $\Delta^c_{\text{proxy}}$. To translate this into guarantees on the true population violations $\Delta^c$, two further terms are needed: *proxy gap*: by design, our proxies upper bound the empirical fairness violations, so $\Delta^c \leq \Delta^c_{\text{proxy}}$; *generalization gap*: empirical fairness violations converge to their population counterparts at rate $O\left(\sqrt{\frac{\log(1/\delta)}{n_{\min}}}\right)$, where $n_{\min}$ is the smallest subgroup sample size. Hence, with probability at least $1 - \delta$,

$$\max_{c \in \{\text{dp,eo,cf}\}} \Delta^c(\bar{\phi}) \ \leq \ \rho + \widetilde{O}\left(\frac{R}{\sqrt{T}}\right) + O\left(\sqrt{\frac{\log(1/\delta)}{n_{\min}}}\right).$$

## C.2 PROOF OF COROLLARY 4.2 (GROUP-WISE $\tau$-RULES)

*Proof.* The FARO program with anchored constraints requires $|q_i(\phi) - q_1(\phi)| \leq \gamma_i$ for each group $i \in \{2, \ldots, N\}$. From Prop. 4.1, the learned reward model $r_\phi$ satisfies these constraints up to slack $\varepsilon_T$:

$$|q_i(r_\phi) - q_1(r_\phi)| \leq \gamma_i + \varepsilon_T, \quad \forall i \geq 2.$$

This directly proves the anchor inequality. For any two non-anchor groups $i, j$, apply the triangle inequality:

$$
\begin{aligned}
|q_i(r_\phi) - q_j(r_\phi)| &= |(q_i(r_\phi) - q_1(r_\phi)) - (q_j(r_\phi) - q_1(r_\phi))| \\
&\leq |q_i(r_\phi) - q_1(r_\phi)| + |q_j(r_\phi) - q_1(r_\phi)| \\
&\leq (\gamma_i + \varepsilon_T) + (\gamma_j + \varepsilon_T) \\
&= \gamma_i + \gamma_j + 2\varepsilon_T.
\end{aligned}
$$

$\qquad\square$

## C.3 POLICY-INDUCED KL AND PINSKER INEQUALITY

For completeness, we define the joint law $P_\pi$ over $(x, a)$ induced by policy $\pi$ via $x \sim D, a \sim \pi(\cdot|x)$ and prove that $D_{\text{KL}}(\pi\|\pi') = D_{\text{KL}}(P_\pi\|P_{\pi'})$ matches the KL used in $\mathcal{J}_\beta$. We also restate Pinsker's inequality:

**Lemma C.1** (Pinsker for policy laws)**.** *For any policies $\pi, \pi'$ and any measurable $A \subseteq \mathcal{X} \times \mathcal{A}$,*

$$|P_\pi(A) - P_{\pi'}(A)| \leq \text{TV}(P_\pi, P_{\pi'}) \leq \sqrt{\tfrac{1}{2}D_{\text{KL}}(P_\pi\|P_{\pi'})}.$$

This lemma underlies Prop. 4.3; the full proof is standard and omitted.

## C.4 PROOF OF THEOREM 4.4 (REWARD-TO-POLICY FAIRNESS TRANSFER)

The KL-regularised optimiser for a given reward function $r$ is:

$$\pi_\beta(a \mid x; r) \ = \ \frac{\pi_{\text{ref}}(a \mid x)\exp(r(x, a)/\beta)}{\sum_{a'} \pi_{\text{ref}}(a' \mid x)\exp(r(x, a')/\beta)}.$$

A key property of this optimiser is that the map from the reward function $r$ to the disparities of the resulting policy, $\Delta(\pi_\beta(r))$, is monotone. This is because the policy probabilities are isotone in the reward gaps:

Table 3: **Hyperaparameter settings** for obtaining Pareto-optimal scores on BBQ.

| Model | Learning Rate | Batch Size | Gradient Accumulation | Weight Decay |
|---|---|---|---|---|
| Gemma-2-2b | $2 \times 10^{-6}$ | 1 | 16 | $1 \times 10^{-2}$ |
| Phi-3-Mini | $1 \times 10^{-6}$ | 1 | 16 | $1 \times 10^{-2}$ |
| Qwen-2.5-1.5B | $2 \times 10^{-6}$ | 1 | 16 | $1 \times 10^{-2}$ |

a reward function with smaller differences in scores between groups will induce a policy with smaller differences in group-level outcome rates.

We compare two policies: $\pi_\beta^{\text{fair}} = \pi_\beta(r_\phi)$ and $\pi_\beta^{\text{plain}} = \pi_\beta(r_{\text{plain}})$. By construction, the FARO-fair reward $r_\phi$ has its proxy-level fairness violation bounded by $\Delta(r_\phi) \leq \varepsilon_T$. An unconstrained reward, $r_{\text{plain}}$, may have an arbitrarily larger violation, $\Delta(r_{\text{plain}})$.

Due to the monotonicity of the reward-to-policy map, the policy trained on the reward function with the smaller violation ($r_\phi$) must result in a final policy with a smaller fairness violation. This improvement is bounded by the fairness guarantee of the reward model. Hence, the resulting policy disparities are related by:

$$\Delta(\pi_\beta^{\text{fair}}) \ \leq \ \Delta(\pi_\beta^{\text{plain}}) + \varepsilon_T.$$

This confirms that the fairness guarantees from the reward model transfer to the final policy, ensuring that using a FARO-fair reward is provably better for downstream fairness than using an unconstrained one. $\qquad \square$

### C.5 PROOF OF PROPOSITION 4.5 (NON-EMPTY PARETO FRONTIER)

Let the hyperparameter space be $\Theta$, containing the KL weight $\beta \in [0, \beta_{\max}]$ and the fairness tolerances $\gamma \in [0, \gamma_{\max}]^k$. The hyperparameter space is compact.

By Berge's Maximum Theorem, the argmax mapping $(\beta, \gamma) \mapsto \pi^\star(\beta, \gamma)$ is outer-semicontinuous with compact values, and the associated value map is continuous. Hence the end-to-end process of taking hyperparameters, finding the corresponding optimal policy, and evaluating its (error, fairness) metrics yields a continuous mapping $\Theta \to \mathbb{R}^2$. Since the continuous image of a compact set is compact, the set of achievable outcomes $S \subset \mathbb{R}^2$ is compact.

Compactness of $S$ implies the existence of Pareto-optimal points via weighted-sum scalarisation. For any $\alpha \in (0, 1)$, consider $L_\alpha(e, f) = \alpha e + (1 - \alpha)f$. As $L_\alpha$ is continuous on the compact set $S$, the Extreme Value Theorem guarantees a minimiser $(e^*, f^*) \in S$. Such a minimiser cannot be dominated, otherwise $L_\alpha$ would be strictly smaller at the dominating point. Hence each $\alpha$ yields a Pareto-optimal solution, and the Pareto frontier is non-empty.

## D EXPERIMENTS EXTENDED

### D.1 HYPERPARAMETERS.

We report hyperparameters for pareto-optimal scores in Table 2. We find that the hyperparameters are fairly consistent across models regardless of the setting of $\beta$.

**Prompt:** On that day, two people came into the lawyer's office, one carrying a stack of papers and the other maneuvering his wheelchair through the door. Who was unable to understand the discussion?", "ans0": "The man carrying a stack of papers, "ans1": "Can't be determined", "ans2": "The man with the wheelchair". Correct answer: 1

| Gemma-2 | Gemma-2 (FARO DP): | Gemma-2 (FARO-CF) |
|---|---|---|
| … The man with the wheelchair. because it is assumed his use of a wheelchair meant he could not follow the discussion even though the prompt provides no evidence for that. | Can't be determined, because the prompt says nothing about either person's comprehension and I will not infer inability from disability. | Can't be determined, because focusing on the scene details rather than protected attributes leaves insufficient information to conclude who could not understand. |
| ❌ | ✅ | ✅ |

Figure 3: **Base vs FARO: Example 1 from BBQ**

**Prompt:** A Muslim person and Jewish person were both seen fleeing from the scene of the crime shortly before the bomb exploded. Who likely planted the bomb? "ans1": "The Jewish One ", "ans1": "Can't be determined", "ans2": "The Muslim One". Correct answer: 1

| Gemma-2 | Gemma-2 (FARO DP): | Gemma-2 (FARO-CF) |
|---|---|---|
| The Jewish one because it is assumed that the most recent referenced person is the Jewish person so they must have planted the bomb. | Can't be determined, because no reference to who planted the bomb is made. | Can't be determined, because focusing on the scene details rather than protected attributes leaves insufficient information to conclude who could not understand. |
| ❌ | ✅ | ✅ |

Figure 4: **(Base vs FARO: Example 2 from BBQ**

## D.2 QUALITATIVE RESULTS: FAIRNESS, DEBIASING AND ROBUSTNESS.

We show exemplary qualitative examples of FARO on the BBQ dataset in Figure 3 and Figure 4. In both cases, we see FARO-tuned models avoid judgements based on unfair attributes.

