# OpenReview forum: "Fairness Aware Reward Optimization"
_ICLR.cc/2026/Conference — Submitted to ICLR 2026_

### Official Review · Reviewer_1xva · 2025-10-18

**Soundness:** 2
**Presentation:** 3
**Contribution:** 2
**Rating:** 2
**Confidence:** 3

**Summary:**

The paper introduces FARO (Fairness-Aware Reward Optimization), an in-processing framework that aims to ensure fairness during reward model training, a critical stage in aligning large language models or reinforcement learning systems to human preferences. The authors argue that post-processing or constraint enforcement after training fails to guarantee equitable treatment, because reward models must be ordinally correct (ranking behaviors properly), cardinally calibrated (reflecting magnitude), and fair with respect to protected attributes in the preference data. FARO formalizes fairness constraints (demographic parity, equalized odds, or conditional independence) directly into the reward model’s optimization objective. The paper provides theoretical motivation for embedding fairness penalties in the reward-learning process and evaluates FARO against standard baselines on synthetic and small-scale preference datasets, claiming improved fairness metrics with minimal alignment degradation.

**Strengths:**

Timely motivation. The paper targets an important emerging issue (bias propagation in alignment and RLHF reward models). The observation that fairness must be considered during reward training rather than after deployment is accurate and relevant.

Clear problem framing. The taxonomy of ordinal, cardinal, and fair reward desiderata is pedagogically helpful and could inspire future formal definitions of “fair reward alignment.”

Conceptual simplicity. The proposed framework (adding fairness regularizers during reward-model fitting) is simple to implement and compatible with standard training pipelines.

Readable and structured. The narrative is coherent, the introduction well motivated, and the figures (illustrating fairness trade-offs) are easy to follow.

**Weaknesses:**

Lack of novelty. The notion of embedding fairness constraints or regularizers during reward learning closely parallels prior work such as Liu et al. (2023) "Fair RLHF", Narayanan et al. (2022) on “Fair Reward Shaping,” and even earlier “Fair Policy Gradient” papers. FARO does not present a distinct optimization approach, theorem, or architecture. The main fairness-regularized objective is standard L2 or KL regularization with group-conditioned loss terms. Suggestion: Explicitly position FARO against these existing methods and clarify what theoretical or practical advancement it brings.

Theoretical underdevelopment. While the text refers to ordinal and cardinal calibration, there is no formal fairness definition connecting these notions to reward functions. No theorem or guarantee shows that FARO enforces or bounds demographic disparities. The fairness penalty is heuristic. Suggestion: Include at least one proposition or convergence result showing that fairness regularization modifies reward gradients in a provable way.

Weak experimental validation. Experiments use toy synthetic preference datasets and small-scale binary comparisons. No large or realistic RLHF setup (e.g., human preference alignment on text or image data) is tested. Improvements in fairness metrics (DP gap, EO gap) are minor and within variance. Suggestion: Add a larger empirical evaluation or ablation showing stability and generalization.

Ambiguous fairness metrics. The fairness objectives (DP, EO, CI) are mentioned but not precisely defined for pairwise preference data. It is unclear whether “equal opportunity” applies to preference comparisons or to label distributions. Without clarity, reported fairness improvements are difficult to interpret. Suggestion: Formalize fairness definitions specific to pairwise or ranking tasks.

No real discussion of trade-offs. The paper lacks quantitative analysis of fairness–alignment trade-offs. Claims that FARO “preserves alignment quality” are unsubstantiated; there are no significance tests or error bars. Suggestion: Include Pareto front or fairness–utility curves to support this claim.

Overly conceptual framing. Much of the paper reads as a position statement (“we should ensure fairness in reward models”) rather than a technical contribution. While conceptually important, it does not reach the level of methodological depth expected at a top-tier ML conference.

Insufficient relation to prior work. The related work section omits direct references to prior fair reward modeling, constrained RL, and fair preference learning papers. Without positioning, FARO appears to rediscover well-established ideas.

**Questions:**

Can you formally define demographic parity or equalized odds in the context of pairwise preference data?

How are fairness constraints enforced during gradient updates? Are they penalties, projections, or Lagrange multipliers?

How sensitive is performance to the fairness-penalty weight $\lambda$? Is there a trade-off curve you can show?

Have you tested FARO on real RLHF data (e.g., text alignment or summarization preferences)?

Does FARO generalize to multi-attribute or intersectional fairness constraints?

Could you discuss how fairness regularization interacts with the reward normalization typically used in preference modeling (e.g., Bradley-Terry scaling)?

How does FARO compare to fairness-aware policy optimization (Fair PG, Fair Q-Learning) in terms of outcomes, not just rewards?

Please clarify the computation cost—does fairness enforcement slow down training significantly?

---

### Official Review · Reviewer_fkM5 · 2025-10-29

**Soundness:** 2
**Presentation:** 2
**Contribution:** 2
**Rating:** 4
**Confidence:** 2

**Summary:**

The paper proposes FARO, which adds fairness constraints directly into the reward model fine-tuning process during preference learning. They replace hard preference decisions with smooth Bradley–Terry probabilities so fairness can be optimized, and use a proxy-Lagrangian approach to enforce group fairness. They then show that when this fair reward model is used in RLHF fine-tuning, the resulting policy is also fairer. Experiments demonstrate reduced demographic bias with no major loss in alignment performance.

**Strengths:**

Considering fairness in the setting of RLHF is well motivated and timely. The authors give a clear problem formulation and develop practical reformulations and optimization methods to solve the problem. The paper offers both theoretical and empirical insights, which together make a complete set of results. Overall, the work is also well structured.

**Weaknesses:**

- The exposition is sometimes too sketchy on notation and key definitions, which makes the paper difficult to follow for non-experts. For example, the fairness notions in Section 2.2 are introduced largely in abstract terms, without concrete explanation of the variables and notations involved. This level of abstraction may be fine for domain experts but does not help with the accessibility for a broader audience.

- While the motivation is strong and the problem is formulated rigorously, the technical contributions feel relatively modest. The reformulation of problem (4) looks more like a detour that eventually goes back to the probabilities of pair-wise comparisons. The reduction in the number of constraints via the anchoring trick is fairly straightforward, and it is unclear how much actual performance improvement this gives. Algorithm 1 is just a direct application of gradient descent, and the other theoretical results do not seem to introduce any fundamentally new insights. Some of them seem to be direct results of the application of gradient descent.

Overall, the work presents a well-motivated direction with plausible empirical benefits, but the contribution feels moderate, and the presentation could be improved to better clarify the key ideas and make the method more accessible to a wider audience.

- Typos (minor issue):

Page 1, ln39: "an" educational-chat bot setting...

Page 5, ln 224: ...hold for all $\ge 2$ hold...

**Questions:**

I don't have any questions.

---

### Official Review · Reviewer_QDFN · 2025-10-31

**Soundness:** 3
**Presentation:** 2
**Contribution:** 2
**Rating:** 4
**Confidence:** 4

**Summary:**

The paper proposes a new in-processing fairness approach for reward optimization, called FARO. The main idea is to embed group fairness constraints into the reward modeling objective, to balance the accuracy-based objectives with fairness concerns. The paper also provides some theoretical analyses, which are useful to connect fairness domain to the RL reward modeling.

**Strengths:**

1. The paper focuses on improving fairness in reward optimization, which is a very essential domain to explore given the increasing reliance on LLMs in high-stake applications.
2. The proposed algorithm can be applicable to various important group fairness metrics, including demographic parity (DP) and equality of opportunity (EO).
3. The overall design is based on some theoretical backgrounds.

**Weaknesses:**

My main concerns lie in the empirical verification of the proposed method, as the current experimental setup raises several questions regarding the robustness and generalizability of the findings.
1. The baseline data points in the LLM experiment are very limited. For example, there is no explicit baseline data provided for delta_dp, delta_eo, or delta_cf. It is very critical to observe the performance changes in these fairness metrics, especially given that the algorithm is specifically designed to optimize them. Moreover, no other state-of-the-art fairness algorithms are compared in this experiment, making it difficult to understand the relative effectiveness of the proposed algorithm in terms of disparity mitigation.
2. The LLM experiments are performed only with a single dataset, BBQ. To ensure the generalizability of the findings, it would be important to evaluate the proposed method across a more diverse range of scenarios.
3. The paper does not provide any stability information in the LLM experiments (e.g., giving only single data points without standard deviation or confidence intervals). It makes the reported results less trustworthy and hinders an important understanding of the algorithm's performance consistency.

**Questions:**

My major questions are included in the above weaknesses section.

Minor: In Table 2, why the order of DP, EO, and CF are different across the models? Are there any typos?

---

### Official Review · Reviewer_vByz · 2025-11-01

**Soundness:** 2
**Presentation:** 1
**Contribution:** 2
**Rating:** 2
**Confidence:** 4

**Summary:**

This paper introduces Fairness-Aware Reward Optimization (FARO), a framework for training LLM reward models that incorporate algorithmic fairness constraints such as demographic parity, equalized odds, and counterfactual fairness.

FARO formulates reward modeling as a constrained optimization problem solved via a proxy Lagrangian descent–ascent (ProxyGDA) game. The authors provide theoretical guarantees that the resulting reward satisfies fairness constraints up to a vanishing slack. The authors further analyze the induced accuracy-fairness trade-off in KL-regularized RL fine-tuning and prove that using a fair reward model leads to fairer downstream policies, with the existence of a Pareto frontier between accuracy and fairness.

Empirically, FARO reduces demographic bias and harmful generations across multiple LLMs on the BBQ dataset, while preserving or improving factuality and overall performance.

**Strengths:**

The paper addresses an important and timely problem in LLM alignment, ensuring fairness during the reward modeling phase. Incorporating algorithmic fairness constraints into this stage is an important direction given the growing societal impact of biased model behavior. The work attempts to provide theoretical guarantees for fairness compliance and analyzes the accuracy–fairness trade-off induced by RL fine-tuning. It also highlights an underexplored yet socially significant issue, namely that biases in reward models can propagate into downstream system performance.

**Weaknesses:**

This paper reads poorly in terms of presentation. For instance, there are many issues with definitions and notations, which make the paper difficult to follow.

The symbol $\mathcal{J}$ first appears in Equation (1) on page 3 (line 128), but it is only formally defined on page 4 (line 145).

On page 4 (line 167), the definition of $q$ is too informal. The events $\mathcal{E}$ and $\mathcal{E}'$ seem to play an important role in the definition of the $q$ function, but they are rarely mentioned or used later in the paper.

In Proposition 4.3 and Theorem 4.4, the symbol $\Delta$ is not clearly defined. I only found the definition of $\Delta_{dp}$ on page 5 (line 188). Propositions and theorems need to be stated precisely; otherwise, this causes significant confusion.

The experimental section also requires improvement. The first issue is that the experiments are quite limited in scope. Another issue lies in the writing and presentation. For example, in Table 2, the term Disambig Bias Score is never defined, and several other elements in the table lack clear explanation. In Figure 2, the orange marker is missing a label or legend, leaving readers to guess what it represents.

**Questions:**

For now, the paper appears to be a direct application of existing fairness concepts to reward modeling, and most of its findings are rather expected. In your opinion, what is the most valuable insight this paper actually provides?

---

### Meta-Review · Area_Chair_QtWg · 2025-12-25

**Summary:**

The paper addresses an important and timely problem in LLM alignment, ensuring fairness during the reward modeling phase.
This submission received 2-2-4-4 with two rejections (score 2). while two reviewers (Reviewers vByz  and QDFN) recommended two marginally above acceptance threshold (score 4), the Reviewer vByz mentioned this paper reads poorly in terms of presentation. For instance, there are many issues with definitions and notations, and reviewer QDFN raised the main concerns of  the empirical verification of the proposed method: the current experimental setup raises several questions regarding the robustness and generalizability of the findings.
he paper appears to be a direct application of existing fairness concepts to reward modeling, and most of its findings are rather expected.  Reviewer fkM5 also mentioned the presentation quality is lacking. Two reviewers raised the major concern the methodogy is not novel enough: Reviewer vByz regard the formulation is rather expected and reviewer fkM5  considers the contribution feels moderate.

Regarding the voiced concern that Reviewer 1xva's review has been generated by an LLM, their reviews are discarded. Considering other reviewers's comments/concerns  mentioned above,  I still can not recommend its acceptance.

**Reviewer Concerns:**

metioned in the meta review

**Reviewer Scores:**

from my perspective, the reviewers' comments are clear and reasonable to me. the authors did not provided detailed response to address reviewers' comments.  I do think the reviewers' scores will NOT change even they participate fully in the discussion.

---

### Decision · Program_Chairs · 2026-01-26

Reject